# Gas Permeability Evolution of Coal with Inclusions under Triaxial Compression-Lab Testing and Numerical Simulations

**DOI:** 10.3390/ma15238567

**Published:** 2022-12-01

**Authors:** Yufeng Zhao, Heinz Konietzky, Thomas Frühwirt, H.W. Zhou

**Affiliations:** 1Research Institute of Petroleum Exploration & Development, PetroChina, Beijing 100083, China; 2Geotechnical Institute, TU Bergakademie Freiberg, 09599 Freiberg, Germany; 3School of Energy and Mining Engineering, China University of Mining and Technology, Beijing 100083, China

**Keywords:** coalbed methane, coal sample, sample reconstruction, triaxial compression test, coupled simulation, discrete element model

## Abstract

Coalbed methane (CBM) exploitation leads to permanent stress redistributions in the coal bodies connected with fracturing processes and permeability changes due to deformation induced internal pore-fracture networks. Gas permeability evolution of coal samples is investigated with a newly developed three-dimensional fluid-mechanical coupled experimental system. X-ray CT is used to investigate the internal structure of the coal samples and delivers the basis to set-up numerical twins. The work focuses on coal samples with inclusions. A novel coupling procedure between two different tools—discontinuum and continuum codes—is established to simulate the permeability evolution. The permeability is related to the crack pattern in general, and crack width in particular. A prediction of permeability is proposed based on fracture distribution and microcrack behavior. The experimental studies validated the coupling approach. Shear fractures cause substantial permeability enhancement. Piecewise relations between permeability and volumetric strain can be used to fit the whole process, where a nonlinear exponential relation is established after the expansion point. The inclusions as important structural characteristics influence this relation significantly.

## 1. Introduction

Coal occupies a major proportion of the energy resource structure of China. Resource exhaustion is verified for shallow coal seams; therefore, exploitation is extended to greater depths. The organic components of coal influence the methane adsorption capacity and diffusion [1,2], while the main flow paths including pores and fractures determine the gas permeability of coal and the production of coalbed methane (CBM) [3,4]. Therefore, the quantification of volume and spatial distribution of pores and fractures in coal are fundamental requirements for CBM reservoir evaluation and gas outburst prediction. X-ray CT became an effective non-destructive method for analyzing internal structures in rocks, especially fractures in coal [5,6,7]. High-resolution lab testing is helpful to develop a deeper understanding of the fluid-mechanical coupled processes at the microscale.

Numerical methods used in rock mechanics can be subdivided into three categories: (1) continuous methods including the finite difference method (FDM), finite element method (FEM) and boundary element method (BEM), (2) discontinuous methods including discrete element method (DEM) and particle method (PM) and (3) mixed methods [8].

According to the task, each method has its own advantages and drawbacks. It is important to choose proper methods for specific cases. Scholars optimized the numerical simulation approaches significantly. For instance, for micro-mechanical models, CT techniques were introduced to reconstruct coal sample models [9] and inhomogeneous granite samples were simulated by numerical models based on Weibull parameter distributions [10]. With the rapid development of computational techniques (hardware and software), discontinuum methods such as the discrete element method [11], and especially particle-based approaches [12,13,14], are widely used in simulating the micromechanical behavior.

The gas permeability evolution is a dynamic process closely related to microscopic damage. Coal is a typical minimally permeable rock material. However, opening and closure, as well as extension of fractures, influence the permeability significantly [15]. Permeability is an inherent property, but the measurement of permeability is influenced not only by the sample itself, but also by the fluid properties and test conditions [16,17]. Gases are compressible, and fluid compression and expansion affect the measurements. The constant head permeability test applying Darcy’s law is the most common steady-state test method. Equation (1) [18], derived from Darcy’s law, gives the corresponding relation for gases, which was used to determine the permeability via hydro-mechanical coupled triaxial laboratory tests.
(1)$$k=\frac{\mu Q_{0}}{A}\frac{2P_{0}H}{\left(P_{u}^{2}-P_{d}^{2}\right)}$$
where, *k* is permeability (m^2^), *μ* is viscosity (Pa·s), *Q*_0_ is volumetric flow rate (m^3^/s) at reference pressure *P*_0_ (Pa), *A* is the cross-sectional area (m^2^), *P_u_* and *P_d_* are upstream and downstream pressures (Pa) applied to ends of the sample and *H* is sample length (m).

The permeability of fractured coal relies on the geometrical parameters of the fracture network, such as orientation, extension, aperture and density of fractures [19]. Mechanical loading leads to stress redistributions inside the sample, which is responsible for microcrack evolution and eventually permeability changes. According to experiments, dilatancy has significant impact on permeability, simply a single fracture and small shear displacement can cause a drastic increase in permeability [20].

Previous simulations were not able to duplicate the permeability evolution based on the fracture evolution at the microscale. This paper presents a coupling between a particle-based approach, which considers the micromechanical damage evolution under loading, and a continuum-based approach to simulate the gas flow. The coupled simulation approach comprises the following steps: (1) CT based analysis of the microstructure; (2) set-up of a particle-based micromechanical model (numerical twin); (3) application of mechanical loading on the particle-based model and observation of damage evolution; (4) set-up of equivalent continuum-based model; (5) transfer of damage pattern from the particle-based model to the continuum-based model; (6) simulation of gas flow in the continuum-based model. The research flowchart is shown in Figure 1.

## 2. CT Image Reconstruction

Five particle-based numerical twins with inclusions (C1, C2, C3, C4 and C5) are generated based on the CT analysis results of coal samples. The generation process to duplicate a numerical model based on the real sample C1 is illustrated in Figure 2 [21]. The components (coal matrix and inclusions) of other models are shown in Table 1 and Figure 3. Particle size was chosen in such a way that the diameter corresponds to the smallest thickness of the inclusions.

The inclusions, represented by red particles, have higher strength than the coal matrix. The inclusions in C1 (shown in Figure 3) are distributed mainly parallel to the axis of the sample. On the contrary, the inclusions in C4 (shown in Figure 3c) are located isolated near the side surface, and the inclusions in C5 are gathered at the top end (shown in Figure 3d). Corresponding numerical twins are generated as a documented example in Figure 4.

## 3. Permeability Test under Triaxial Compression

The permeability evolution of coal samples is investigated via triaxial fluid-mechanical coupled tests (Karman type) arranged for steady-state flow. The tests are conducted at a room temperature of 20 ± 3 °C. A stiff servo-controlled loading device is used to apply axial load and confining pressure. The cylindrical sample has been sheathed by a polyvinyl chloride (PVC) core sleeve. Sample and axial platens are isolated from the confining fluid by the core sleeve. The gas used in this experimental research is pure nitrogen. Its viscosity is 1.78 × 10^−5^ Pa·s at laboratory temperature. The gas migrates through the sample without leakage. The system is supported by professional control software packages, such as Test-Star-II Control System and Test-Ware from MTS. The entire set-up of the test is shown in Figure 5.

Compressed nitrogen is provided by a gas tank. The flow rate is controlled by the outlet control valve CV1. The input upstream pressure is adjusted to maintain a constant value. Plug valve PV1 is directly connected to the pressure transmitter PT1 and the sample cell. PT1 is installed to monitor the upstream pressure. Nitrogen gathered at the upstream end is pushed to flow through the coal sample. Pressure transmitter PT2 is installed at the downstream end. PT2 is used to monitor the downstream pressure. The function of plug valves PV1 and PV2 is to guarantee the safety of the sample. Gas flow control valve CV2 and a third pressure transmitter PT3 are attached following PV2. CV2 is used to adjust the downstream pressure. A filter unit is connected to relieve the pressure and clean the gas. Temperature sensor TT1 is to monitor the gas temperature at the outlet. MFC1 and MFC2 are two mass flow controllers. The measurement range of MFC1 is 0–200 cm^3^/min with high accuracy, and the measurement range of MFC2 is 0–1000 cm^3^/min with lower accuracy. At the early stage of the test, by operating the three-way plug valve PV3, MFC1 is used to monitor the flow rate, while MFC2 acts as backup (standby monitoring). If the range of MFC1 is exceeded, MFC2 is activated. The oil syringe pump is connected to the triaxial cell with the plug valve PV4.

The downstream pressure is set to 0.2 MPa. Therefore, the pressure difference is 0.8 MPa. The gas pressure is set always lower than the confining pressure. Exemplary sample C1 under 2.5 MPa confining pressure is used to describe the loading path. Axial loading and confining pressures are raised to 2.5 MPa first (isotropic compression), then the upstream gas pressure is increased gradually to 1 MPa. After about 7 h, the downstream flow rate is detected. By adjusting CV2, downstream pressure is set to 0.2 MPa. After reaching steady flow state, the upper plate is moved at 8 × 10^−5^ mm/s. This process is conducted in several steps: the loading lasts for 750 s followed by a pause, and is then continued. The gas seepage reached stable states in each loading stage. Flow rates, gas pressures as well as mechanical pressures and deformations are continuously monitored to determine the permeability evolution as function of mechanical loading.

## 4. Fluid-Mechanical Coupled Numerical Simulations

The overall permeability calculated by total flow rate and entire cross-section area of the sample is much lower than the permeability of the fractured region. Therefore, the consideration of the evolution of pronounced flow channels is important to get a detailed understanding of the fluid-mechanical coupled behavior of coal, especially if inclusions are present. The combination of continuum- and discontinuum-based numerical simulation methods is a promising way to get information about that process at the microscale. Figure 6 illustrates a specific coupling scheme between the discontinuum code PFC^3D^ [22] and the continuum code FLAC^3D^ [20]. This method is developed to investigate the fluid-mechanical damage evolution of coal samples with inclusions under triaxial compression.

Both models are set-up in parallel by duplicating the microstructure obtained from CT. Size of particles (diameter) corresponds to size of zones (edge length) in the continuum-based model. PFC^3D^ is used to simulate the microcrack/microfracture evolution. FLAC^3D^ is used to simulate the gas flow. At predefined time step intervals, crack data are transferred from PFC^3D^ to FLAC^3D^. Corresponding zones in FLAC^3D^ are assigned with hydraulic parameters based on the crack data and flow-only numerical simulations are performed to obtain the permeability evolution (one-way coupling).

In this paper, two terms “crack” and “fracture” are used. “Crack” corresponds to one broken bond generated between two particles in the PFC^3D^ model. An assembly of connected “cracks” is marked as a “fracture”. The crack size (length) is limited to the size of particles or zones. On the contrary, the size of a fracture depends on the number of connected cracks, the length of one fracture may cover the length of the whole sample.

### 4.1. Numerical Modeling

The reconstructed discontinuum model is assigned with linear parallel bonds between the particles. The numerical simulations are performed in a strain-controlled mode by specifying constant velocities at the top and bottom walls to create a three-dimensional stress state inside the sample in analogy to the corresponding laboratory tests. A small friction coefficient of 0.1 is assigned to the interfaces between loading walls and sample model, respectively. A new approach is developed and applied to simulate the confinement [21]. The flexible membrane is duplicated by wall elements distributed around the sample to provide the confining stress (Figure 7). A servo mechanism acts on each wall element to simulate the confinement with desired constant stress. The simulation duplicates the lab test procedure. The simulation starts with isotropic compression until the desired confinement is reached. Then, the confinement is kept constant, and a constant velocity is applied in vertical direction until peak stress is reached, then further until the residual vertical stress decreases to 60% of the peak stress.

The data exported from the CT analysis is voxel based. The zone elements in FLAC^3D^ are generated in a hexahedral pattern. The final mesh for the continuum model is defined by octagonal meshing in one plane and extrusion into the third dimension.

Crack data, including type, position, radius, normal direction and aperture, are exported from the particle-based model via ASCII files. Each crack covers a circular region with thickness in the continuum model. By calculating the distances from nearby zone centers to the crack center, each crack can be represented by one or several zone elements. New fluid-mechanical parameters are assigned to these zones. Figure 8 illustrates in principle how grouped zones represent fractures with different orientations.

An own-developed script is used to translate the crack characteristics into zone properties. As an example, Figure 9 shows four types of cracks. The model contains 240,000 small zones divided into six element types (two intact matrix types for coal and inclusions and four crack types: matrix shear (fracsm) and matrix tensile (fractm) cracks as well as inclusion shear (fracsi) and inclusion tensile (fracti) cracks). The cracks generated between matrix and inclusion are also included in the inclusion cracks.

Gas flow simulations are conducted for specific states of triaxial compression to replicate laboratory tests. The permeability of the sample model is calculated based on Darcy’s law.

### 4.2. Analysis of Numerical Simulations

The linear parallel bond model, which considers elastic normal and shear stiffnesses as well as cohesion, tensile strength and friction, is used in this research.

The micromechanical parameters of the contacts are calibrated. Particle number (*N*), size (*r_p_*) and porosity (*ρ_p_*) are carefully chosen for modeling. Parallel bond effective modulus (*E*) follows a linear relationship with Young’s modulus. The ratio of normal to shear stiffness (*K**) and the macroscopic Poisson’s ratio show a logarithmic relationship in the elastic stage. The ratio of tensile strength ($\bar{\sigma }$*_t_*) to cohesion ($\bar{c}$) also influences the deformation pattern. The friction angle (*Φ*) affects the ratio of generated tensile to shear cracks. Radius multiplier ($\bar{\lambda }$) is set by default. Contact gap (*g_c_*) is set to detect the bonding condition. Some parameters have cross effects with each other on the macroscopic parameters, such as Poisson’s ratio, peak strength and failure pattern. The finally applied parameters as given in Table 2 are validated by pure mechanical triaxial tests. These simulations are performed with the new flexible wall confining approach [21]. Axial loading and confinements are applied by wall movements.

The crucial point of the simulation is the coupling between crack/fracture propagation and permeability evolution. Theoretical relationships are developed for a single crack characterized by type, size and aperture, and corresponding hydraulic attributes such as porosity and permeability based on previous works [23,24]. The permeability k of the entire sample model at a given loading state is a combination of the permeability of the initial sample (*k*_0_) and the changes caused by cracks/fractures (Δ*k*). The cracks are classified as shear cracks and tensile cracks. These two types have different effects on the hydraulic properties.

For a given crack/fracture aperture, flow rate is proportional to pressure difference [25]. Based on the cubic law, under laminar flow conditions fluid flow rate through a narrow channel can be described by the following equation:(2)$$q=\frac{W^{3}\Delta Pl}{12\mu L}$$
where *W* is the crack width (m); *q* is the gas flow rate through the crack (m^3^/s); Δ*P* is the pressure difference over the extended crack length (Pa); *l* is the length of the crack segment observed on one zone element surface (m); *μ* is the gas viscosity (Pa s); *L* is the side length of the zone in flow direction (m). In the continuum model, each zone is nearly a cube, so the value of *l* is literally equal to the value of *L*. The cross-section of the crack is simplified to be of rectangular shape. From Equation (2) and Darcy’s law, the permeability *k_f_* of a crack inside a zone is defined by the following equation:(3)$$k_{f}=\frac{W^{2}}{12}$$

Shearing of non-planar cracks/fractures is related to dilation, which results in an aperture increase. A given increment in shear displacement (Δ*δ*) leads to a positive change in aperture (Δ*W*), according to previous research [26]. This change can be calculated based on the tangent of the dilation angle (*ψ*):(4)ΔW=Δδtanψ

The particles are in hexagonal close-packed structure. Four initial adjacent particles (P1, P2, P3 and P4) form a regular tetrahedron structure as shown in Figure 10. The mechanism of crack opening calculation is illustrated separately for tensile and shear cracks. The stress-induced breakage of bonds leads to a rearrangement of the particles. The tensile normal displacements between two particle layers follow the relationship shown in Equation (5) and illustrated in Figure 11.
(5)$$W_{T}=D_{T}-D_{i}=\sqrt{{\left(2r_{p}+D_{ap}\right)}^{2}-\frac{4}{3}r_{p}^{2}}-\frac{2\sqrt{6}}{3}r_{p}$$
where *D_T_* is the new distance from P4 to plane (P1, P2, P3); *D_i_* is the initial distance from P4 to plane (P1, P2, P3); *W_T_* is the tensile crack width; *D_ap_* is the particle surface aperture caused by tensile cracking; *r_p_* is the radius of the particles.

Shear displacement may occur along the plane (C1, C2, C3). Dilation angles can be derived as described by Equations (6) and (7) (see also Figure 12).
(6)$$W_{S1}=D_{S1}-D_{i}=\left(\sqrt{3}-\frac{2\sqrt{6}}{3}\right)r_{p}=\Delta \delta_{1}\tan\psi_{1}$$
(7)$$W_{S2}=D_{S2}-D_{i}=\left(2-\frac{2\sqrt{6}}{3}\right)r_{p}=\Delta \delta_{2}\tan\psi_{2}$$
where shear displacements Δ*δ*_1_ and Δ*δ*_2_ are calculated considering different shear orientations; *D_S_*_1_ and *D_S_*_2_ are distances from C4 to plane (C1, C2, C3) after shearing; *W_S_*_1_ and *W_S_*_2_ are the shear crack induced widths. The dominant shear dilation angle *ψ*_2_ is chosen to calculate the crack width according to Equation (7). It is assumed that there is only one crack in each zone. Overall permeability (*k_z_*) of one zone is the sum of two parts, as given by the following equation:(8)$$k_{z}=ak_{f}+\left(1-a\right)k_{i}$$
where *k_f_* is the permeability of the fracture (m^2^); *k_i_* is initial matrix permeability of the zone (m^2^); *a* is area contribution coefficient. The area contribution coefficient (*a*) is calculated considering the ratio of cross-section area. Permeability *k* of one unit is obtained as follows:(9)$$k=\frac{W^{3}}{12L}+\left(1-\frac{W}{L}\right)k_{i}$$

The coalescence of cracked zones contributes to local and overall gas permeability. The permeability is achieved for tensile and shear cracks by substituting *W_T_* and *W_S_* into Equation (9), respectively. The crack width has a major influence on permeability (see first component in Equation (9)). The initial zone permeability has a minor influence, but the value of the second term cannot be neglected when considering the corresponding closed crack.

According to the bond-deformability criterion, a default crack width *W_as_* of 0.05 mm is set to newly generated cracks in FLAC^3D^. Theoretical volumetric strain *ε_VT_* of the model is accumulated. On the other hand, the volumetric strain *ε_VP_* is calculated by the PFC^3D^ model. A scaling coefficient *c_bl_* is applied to individual crack widths by comparing *ε_VT_* and *ε_VP_*. As shown in Equation (10), the coefficient *c_bl_* is assigned to determine the permeability *k_cl_* of new cracked zones.
(10)$$k_{cl}=\frac{{\left(c_{bl}\times W_{as}\right)}^{3}}{12L}+\left(1-\frac{c_{bl}\times W_{as}}{L}\right)k_{i}$$

Based on the previous analysis, the maximum derived width caused by dilation is about 10 times the residual width, which has been proven by references [27].

Based on the given equations related to aperture and crack type, hydraulic parameters, including permeability and porosity, are assigned to the cracked zones. The mechanical and hydraulic parameters for hydraulic simulation are given in Table 3. Density (*ρ*), elastic modulus (*E*), tension limit (*σ^t^*), cohesion (*C*), dilation angle (*ψ*), friction angle (*θ*) and Poisson’s ratio (*ν*) are calibrated by comparing with UCS results obtained by laboratory testing. Intrinsic permeability (*k_a_*), configured permeability coefficient (*K*) and porosity (*φ*) are calibrated by seepage tests.

## 5. Results and Discussions

### 5.1. Laboratory Tests

Five coal samples of a standard size are tested in the laboratory with confining pressures of 2.5 MPa, 5.0 MPa and 7.5 MPa, respectively. Permeability–strain curves and stress–strain curves are obtained. Physical dimensions, confinement settings and peak permeability are shown in Table 4.

The tests indicate a complicated process with axial stress variation. According to the stress–strain curve of sample C1 (Figure 13) as an example, the permeability–strain curve is lagging behind the stress–strain curve. The seepage process (permeability curve is plotted on a logarithmic scale to illustrate the subtle changes) inside the coal sample goes through the following five stages:Initial compaction stage (I). Permeability decreases linearly with increasing axial loading. Internal voids close under compression, so the flow rate decreases due to the shut-down of seepage paths. However, a general permeability decrease is not significant. When the axial differential stress reaches 8.78 MPa, permeability decreases to a minimal level, and the corresponding stress point is defined as σm. It is difficult to observe any gas seepage.Linear elastic deformation stage (II). Beyond the stress point σm, permeability increases moderately. The flow rate increases significantly following the stress–strain behavior. In this stage, initial pores grow and new microcracks are generated.Nonlinear deformation and peak strength stage (III). With continuous increase of axial stress, cracks become wider and coalesce. Macroscopic fractures begin to appear and permeability increases drastically.Strain softening stage (IV). After the peak differential stress (28 MPa), permeability increases continuously with axial strain. Such a scenario lasts until differential stress decreases to 22.74 MPa. Then, permeability and flow rate reach peak values and maintain constant. Fracture apertures increase, fractures coalesce and the fracture network is fully developed.Residual stress stage (V). Ongoing deformations cause abrasion on fracture surfaces. The apertures decrease with ongoing decrease of surface roughness. To some extent compaction appears and permeability is decreasing. This decrease is more pronounced at higher confining pressures.

When σm is reached, the seepage reaches the minimum level. Beyond σm, the seepage changes qualitatively and quantitatively.

Due to restricted sensitivity of the monitoring devices, initial flow rates are difficult to detect. The lowest threshold is 1.5 sccm (standard cubic centimeters per minute), which is equal to 2.5 × 10^−8^ m^3^/s. It is possible to detect σm for sample C1. Another stress point called *σ_d_* is defined when the flow rate is detected for the first time during the test. The corresponding deviatoric strain at stress point *σ_d_* is *ε_d_* (detection point). *ε_d_* is an important critical point to describe the permeability evolution trend.

The permeability evolution for different samples under various confining pressures is shown in Figure 14. The corresponding strain value at peak permeability is defined as *ε_p_* (peak point).

The permeability tends to decrease at a low confining pressure, and the extremely low flow rates at high confining pressure cannot be detected in the early stage due to limits of the measurement devices. The permeability evolution of sample C1 is typical (see Figure 14a). Stress points *σ_d_* are recorded before peak strength for the confining pressures of 2.5 and 5.0 MPa is reached. With 7.5 MPa confining pressure (samples C4), flow rate is detected for the first time beyond peak differential stress.

By analyzing the results, confinements are proven to restrict the permeability evolution. Confining pressures were applied perpendicular to the general flow direction, and fractures are generated predominantly parallel to this axis, and were closed during the first phase. By comparing the models with different confining pressures, the following is observed: before the peak strength is reached, permeability is reduced by more than one order of magnitude when confining pressure changes from 2.5 to 7.5 MPa. In post-peak stages, permeability varies in a much wider range at low confining pressure, sometimes permeability values maintain stable with a minor increase (see sample C1 shown in Figure 14a), or reduce to one third of the peak value (see sample C11 shown in Figure 14b). The post-peak permeability is more sensitive at low confining pressure. Before peak strength is reached, axial stress restricts the seepage to a small extent. Beyond the peak strength, applied axial stress is able to further increase the permeability at low confinement. On the other hand, axial stress causes more constraints on flow channels at high confining pressure. For sample C12 with a confining pressure of 7.5 MPa (Figure 14e), a relatively high permeability is detected at the initial stage because of its internal structural features.

It is believed that the permeability decreases when stresses are compressive on the fracture. On the other hand, the permeability increases significantly when stresses induce tensile/shear displacements on the cracks/fractures. Compared with flow through macroscopic fractures, the permeability of coal for fluids (gas or liquid) is extremely small.

Results also show that peak differential stress and peak permeability are closely related to confining pressure. The relations between peak differential stresses and peak permeability versus confining pressure are shown in Figure 15. Peak differential stresses and peak permeability are predictable for specific confining pressures.

Linear fitting for peak differential stress and peak permeability as function of confining pressure results in Equations (11) and (12), respectively (see Figure 15).
(11)$$\sigma_{P}=18.3432+3.325\cdot \sigma_{3}\left(R^{2}=0.875\right)$$
(12)$$k_{P}=3.56\times 10^{-15}-4.31\times 10^{-16}\cdot \sigma_{3}\left(R^{2}=0.905\right)$$
in which, *σ_P_* is peak differential stress (MPa); *k_P_* is peak permeability (m^2^); *σ*_3_ is confining pressure (MPa).

### 5.2. Numerical Simulations

The hydro-mechanical coupled seepage process is investigated by numerical simulations. For each considered loading stage the simulations are performed until a steady-state flow rate is achieved, and the flow rate is used to calculate the overall permeability.

The pore pressure shows some local variations as documented in exemplary Figure 16 caused by the heterogeneity of the sample, especially the distribution of cracks and their specific hydraulic properties.

The numerical model C1 under 2.5 MPa confining pressure is illustrated as an example. The observed phenomena are further described in detail (see Figure 17). The numerical model allows—in addition to the lab testing—a detailed analysis of the hydro-mechanical coupled behavior at the microscale. Special attention is paid to flow paths formed by hydraulic active cracks and micro-fractures.

In the early loading stage, the permeability remains at a very low value due to two reasons. Firstly, the number of cracks increases slowly, where only 34 new cracks are observed before the axial strain reaches 0.01. Secondly, new microcracks are generated isolated in the sample, and no new flow paths are formed. The existing cracks and fractures remain closed and aperture of opening cracks/fractures become smaller.

When approximately 30% of peak differential stress is reached, the crack number increases drastically, which leads to increasing permeability. Until peak strength (axial strain of about 0.014), the total crack number increases to more than 600. Among them, 22 cracks are detected with noticeable apertures (width larger than 0.05 mm). Significant enhancement of permeability occurs locally, when individual cracks form vertical fractures. Finally, end-to-end flow paths are formed, and flow rate, as well as corresponding permeability, reaches peak values. In the post-failure region, new fractures occur continuously and shear cracks become dominating, but the opening stays restricted by the confinement. As shown in exemplary Figure 17, the overall permeability of the sample C1 remains at a high level after the axial strain reaches 0.025. Please note that we have not expected a close agreement between laboratory tests and numerical simulation results due to the complex inhomogeneity and the restricted resolution of the numerical sample.

In general, permeability *k* for brittle materials is related to volumetric strain *ε_V_* [13]. According to the permeability evolution trends as function of axial stress–strain relations, the permeability evolution can be divided into five stages as shown in Figure 18. In the first stage (A–B), both permeability and volumetric strain decrease with time. The initial cracks become closed at the beginning of this stage, followed by further closure of microscopic pores. In the second stage (B–C), the permeability tends to be stable at a very low level connected with closure of pre-existing pores and cracks, only a few new microcracks are generated. A dynamic balance is reached in this stage. In the third stage (C–D), volumetric strain turns from decreasing to increasing, the sample goes into the volumetric expansion phase (a corresponding expansion point can be noticed), and the cracks start to emerge, expand and connect with each other. Additionally, in this stage the permeability begins to increase, and the axial stress increases gradually up to the peak value. In the fourth stage (D–E), cracks are connected extensively and macroscopic fractures are formed while the axial stress decreases drastically, thus, the sample is in a failure state. The fifth stage (E–F) is the post-failure stage, the volumetric strain increases continuously, but the increase rate slows down. The permeability evolution in this stage shows different types as mentioned before. Figure 19 documents the exemplary evolution of mechanical and hydraulic parameters during mechanical loading up to the post-failure region as obtained by the numerical simulations. The simulation results of the other samples are incorporated in Figure 20 and Figure 21.

The numerical simulation reveals some consistent characteristics considering that the corresponding deviatoric strain of expansion point is defined as *ε_E_*. Before the expansion point *ε_E_*, the permeability *k* is either constant or slightly increasing. The permeability *k* shows an exponential behavior with respect to the volumetric strain *ε_V_* beyond the expansion point *ε_E_*. Only one exception (sample C4) is observed with a linear trend. The point *ε_E_* occurs slightly before peak stress is reached. Differential stress and volumetric strain *ε_V_* show a smooth evolution, but also significant differences when considering different samples. Permeability evolution after the expansion point shows a clear trend, but with remarkable fluctuations. Both *ε_E_* and *ε_d_* are strongly related to the confining pressure as shown in Figure 20.

The linear fitting of *ε_E_* versus confining pressure results in Equation (13) (see also Figure 20).
(13)$$\varepsilon_{E}=3.46\times 10^{-4}+3.08\times 10^{-3}\cdot \sigma_{3}\left(R^{2}=0.9586\right)$$

The numerical simulations show some remarkable scatter due to the heterogeneity of the samples, although the general trend becomes obvious (see Figure 21). The obtained fitting curves may not be able to represent the behavior of individual samples, but deliver reasonable overall trends. The sample volume at the expansion point is much lower with higher confining pressure. By analyzing all simulation results, the corresponding volumetric strains for the expansion points for confining pressures of 2.5, 5.0 and 7.5 MPa are −0.49%, −0.77% and −0.92%, respectively.

Nonlinear exponential trends for permeability increase with volumetric strain are obtained for the expansion stage (see Equation (14) and Table 5).
(14)$$k=A\cdot \exp\left(\frac{\varepsilon_{V}}{C}\right)+B$$

After considering the variations of *A*, *B* and *C* with *σ*_3_, Equations (15)–(17) are obtained.
(15)$$A=-5.1811\times 10^{-16}+5.9618\times 10^{-17}\cdot \sigma_{3}$$
(16)$$B=1.0888\times 10^{-14}\cdot \exp\left(\frac{\sigma_{3}}{1.2992}\right)+7.6229\times 10^{-16}$$
(17)$$C=0.0154\cdot \exp\left(\frac{\sigma_{3}}{2.0408}\right)-0.0083$$

Equation (14) can be used to predict the permeability based on volumetric strain considering confining stresses between about 1.5 and 8.5 MPa, which covers typical stresses in coal repositories.

## 6. Conclusions

The evolution of coal permeability under different stress states has been investigated via HM-coupled conventional triaxial compression tests. The evolution process is described by five stages in terms of permeability and deformation. Peak permeability occurs after peak differential stress. Peak differential stress and peak permeability depend on confining pressure.

A coupling method between PFC^3D^ and FLAC^3D^ is effective in simulating the permeability of damaged coal samples also considering inclusions. A novel approach for defining the hydraulic properties is proposed based on crack behavior and fracture distribution. The properties of both tensile and shear fractures are derived separately based on the mechanical interaction of particles.

Permeability and volumetric strain show a good nonlinear exponential relation after the expansion point *ε_E_*. Derived functions fit the whole process and the expansion point *ε_E_* is regarded as the critical point. In general, the permeability increases drastically after the sample reached the detection points *ε_d_* (when flow rates are detected for the first time or recovered from minimum states). The structural characteristics affect this relation significantly. The simulation results provide guidance for practical gas flow prediction. Based on the experimental evidences, this micromechanical based modeling approach is applicable for the description of induced anisotropic damage and permeability evolution.

## Figures and Tables

**Figure 1 materials-15-08567-f001:**
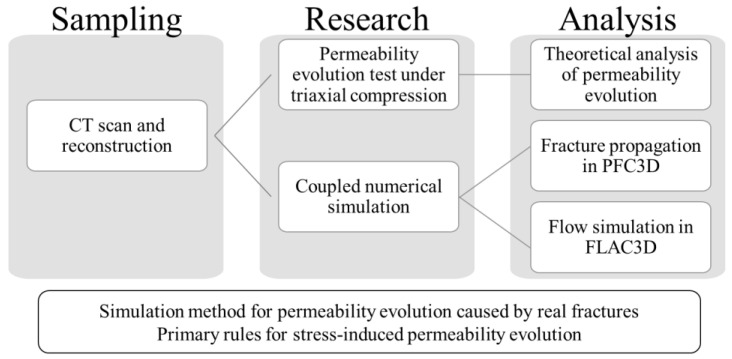
Flowchart illustrating the research strategy.

**Figure 2 materials-15-08567-f002:**
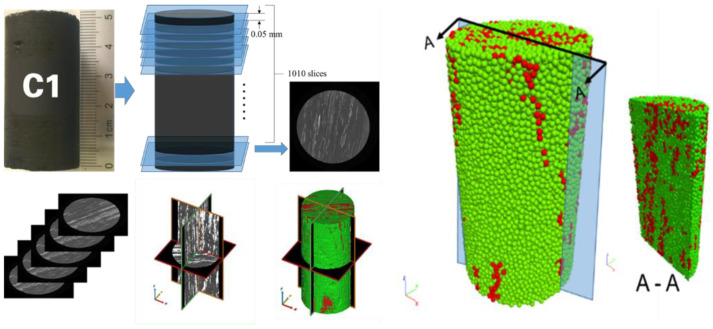
Complete scanning procedure and corresponding reconstructed 3D model of sample C1 (Internal structure shown in A-A cross-section; green: coal matrix, red: inclusions).

**Figure 3 materials-15-08567-f003:**
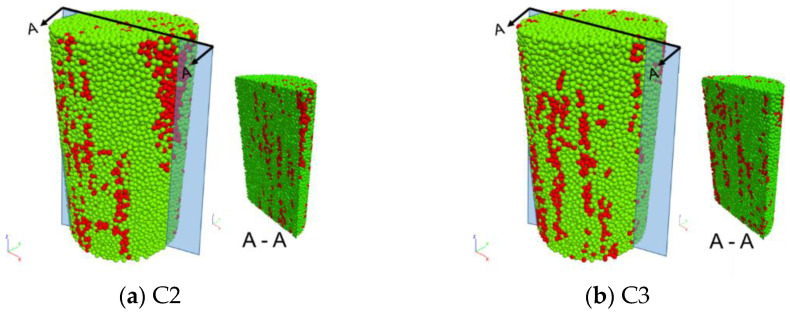
Reconstructed particle-based numerical models of coal samples (green: coal matrix, red: inclusions).

**Figure 4 materials-15-08567-f004:**
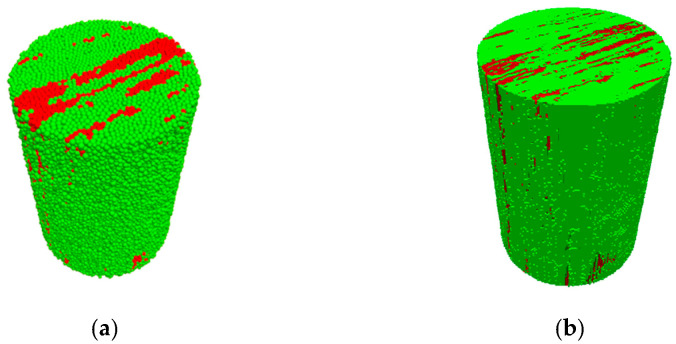
Particle-based (**a**) and corresponding continuum-based (**b**) model C1 (green: matrix; red: inclusions; holes: pores).

**Figure 5 materials-15-08567-f005:**
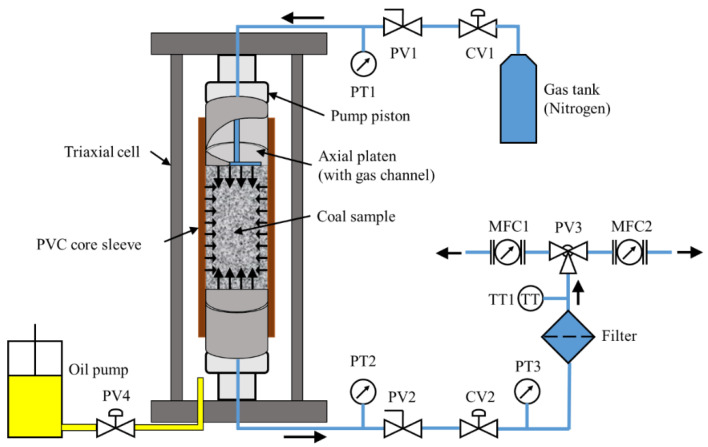
Schematic of the experimental set-up.

**Figure 6 materials-15-08567-f006:**
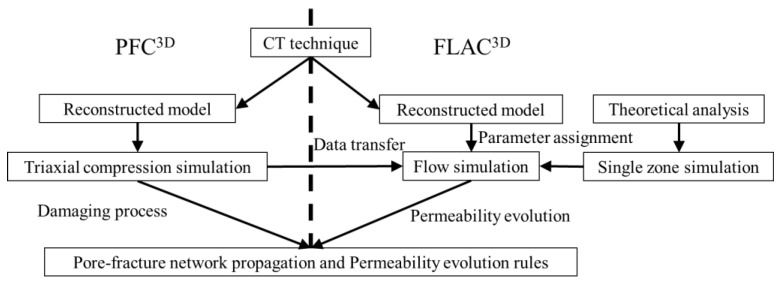
Flowchart illustrating the specific coupling between PFC^3D^ and FLAC^3D^.

**Figure 7 materials-15-08567-f007:**
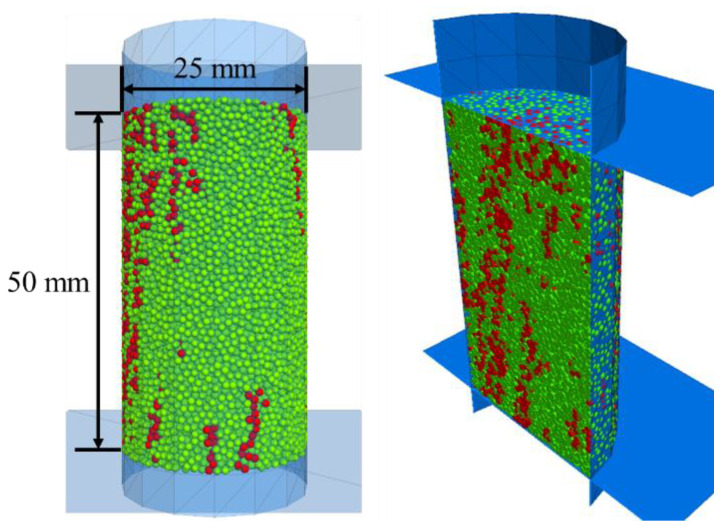
Reconstructed sample with flexible membrane as outer vertical boundary.

**Figure 8 materials-15-08567-f008:**
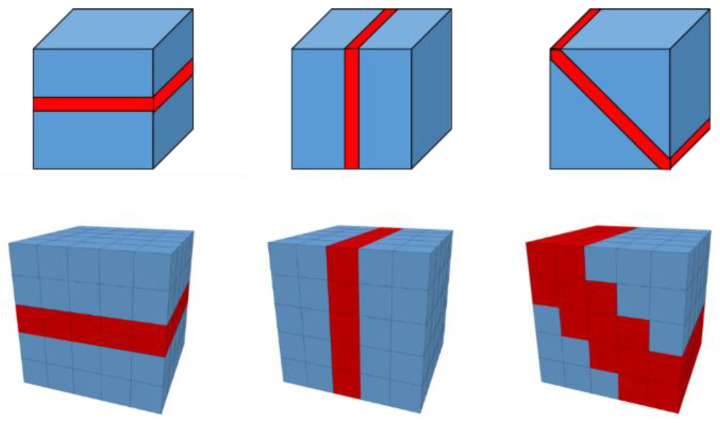
Schematic to illustrate how fractures in principle are represented by zones in the continuum-based model (above: idealized fractures with different orientations, below: representations inside the continuum-based model).

**Figure 9 materials-15-08567-f009:**
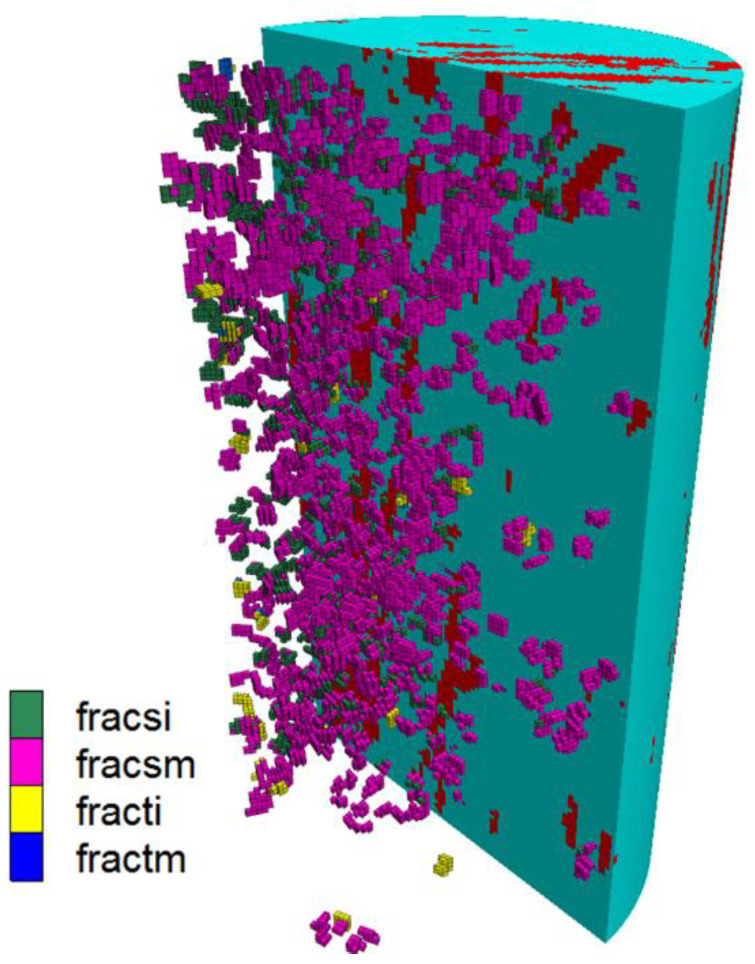
Illustration of a continuum model for a certain triaxial loading stage with four crack types imported from the discontinuum model with the same loading stage.

**Figure 10 materials-15-08567-f010:**
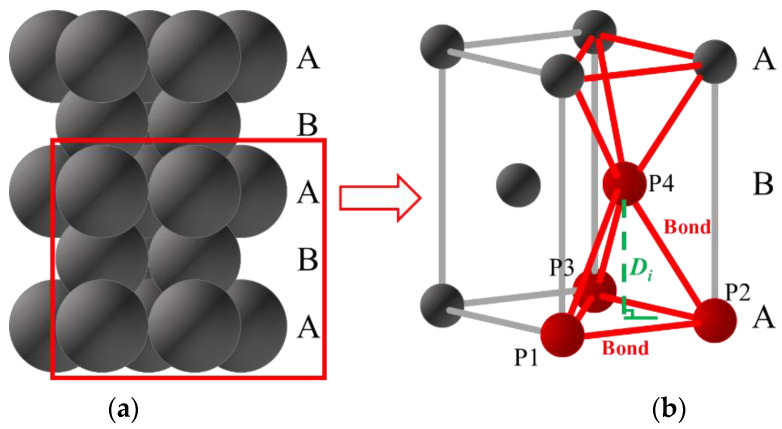
Hexagonal close-packed structure. (**a**) stack of particles in PFC^3D^ (**b**) simplified particle-bond model.

**Figure 11 materials-15-08567-f011:**
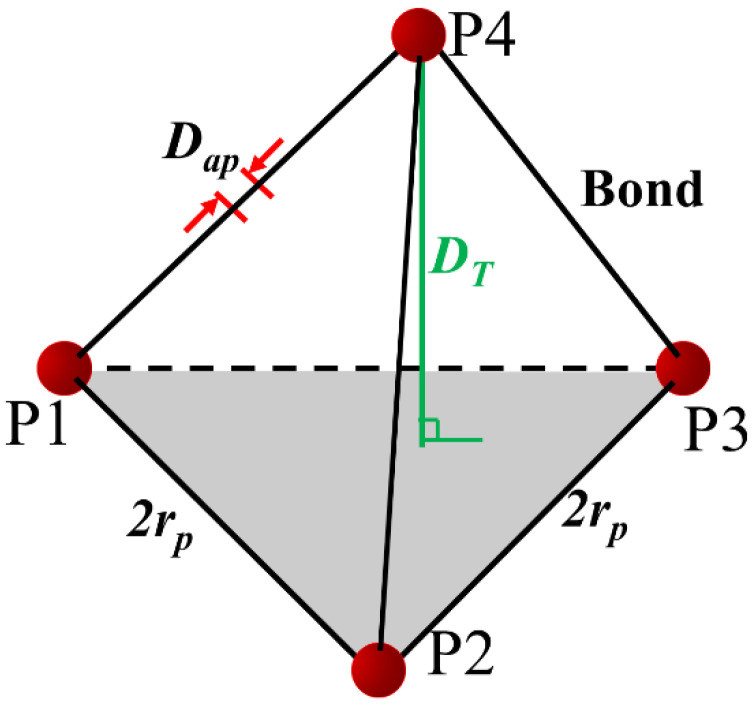
Illustration of the relation between crack width and direct contact aperture.

**Figure 12 materials-15-08567-f012:**
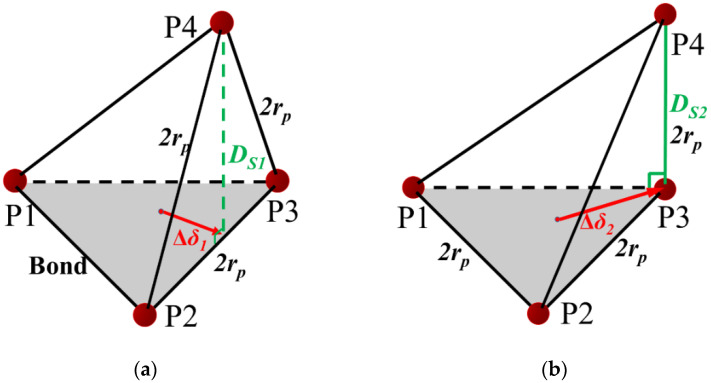
Illustration of the relation between crack width and shear displacement. Shear forms shown in (**a**) and (**b**) are corresponded to Equations (6) and (7) respectively.

**Figure 13 materials-15-08567-f013:**
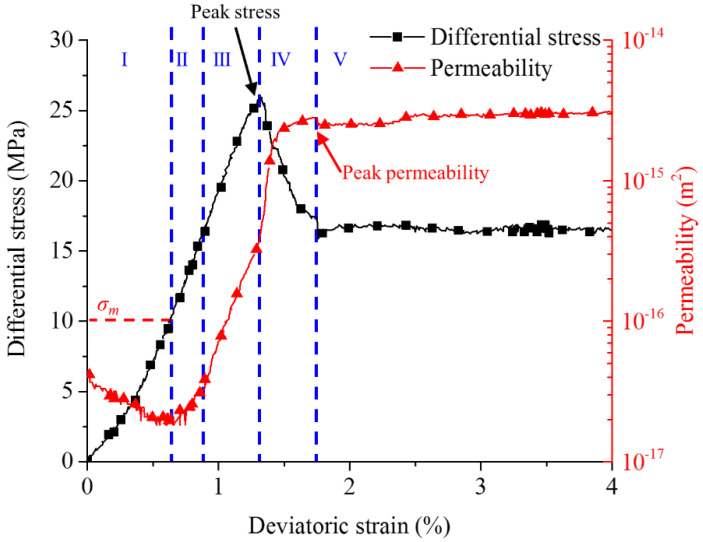
Differential stress (axial stress minus circumferential stress) and permeability evolution versus axial strain for sample C1 (with a confining pressure of 2.5 MPa).

**Figure 14 materials-15-08567-f014:**
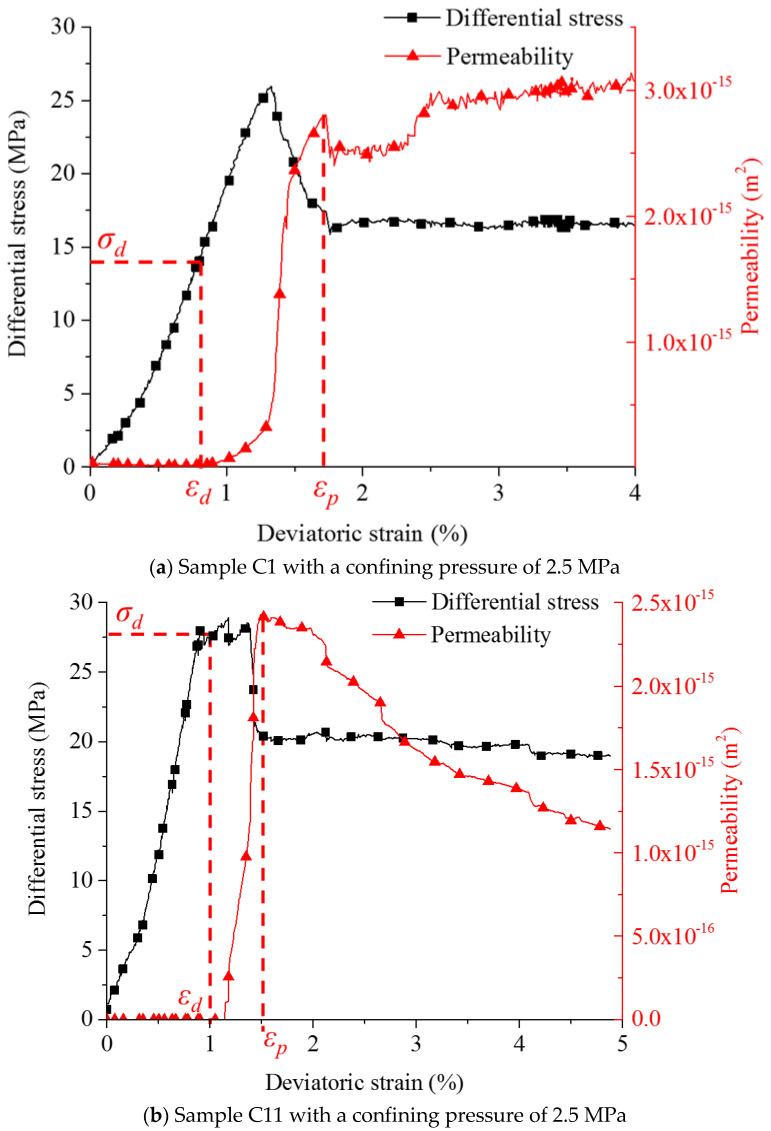
Permeability evolution and complete stress–strain curves for triaxial compression tests.

**Figure 15 materials-15-08567-f015:**
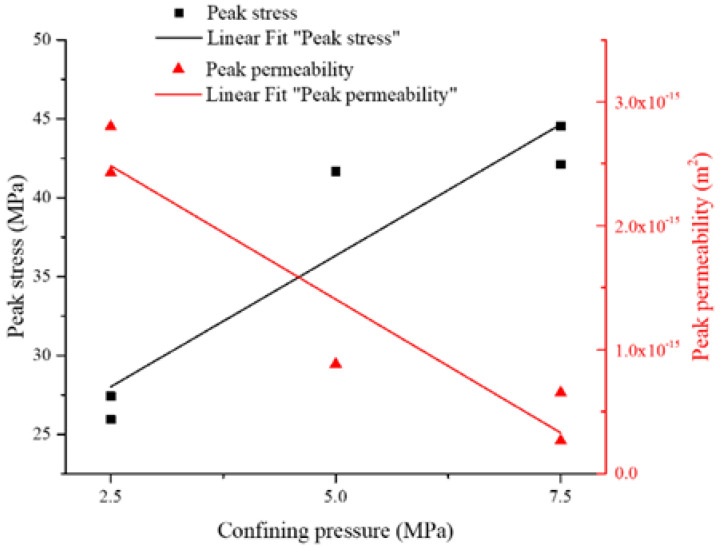
Peak differential stress and peak permeability versus confining pressure.

**Figure 16 materials-15-08567-f016:**
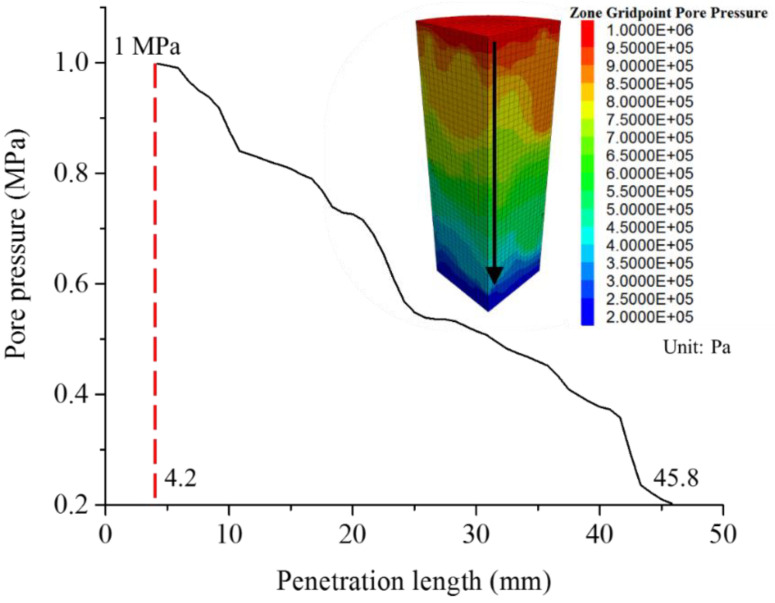
Pore pressure along vertical axis for model C1 (at axial strain of 0.0132, stationary phase) and pore pressure distribution [Pa].

**Figure 17 materials-15-08567-f017:**
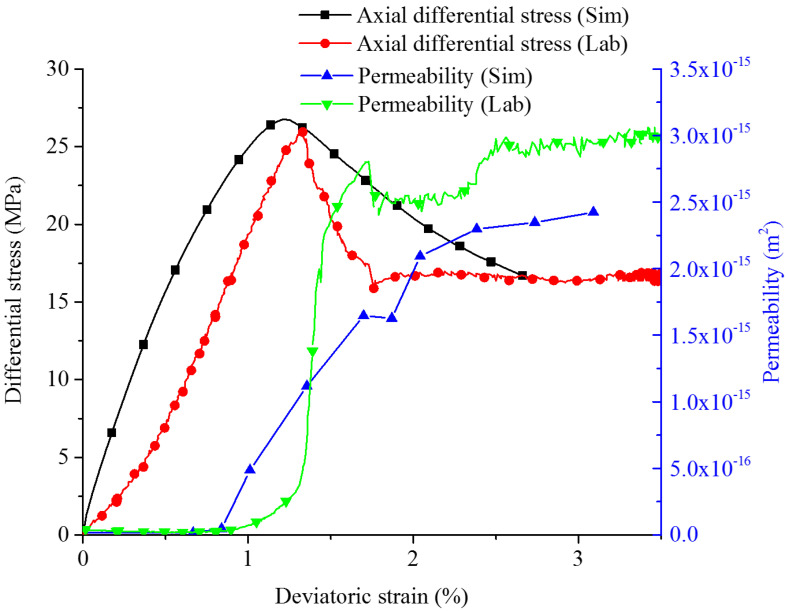
Stress–strain curves and permeability evolution (lab vs. simulation) for sample C1 for 2.5 MPa confining pressure (stationary phase).

**Figure 18 materials-15-08567-f018:**
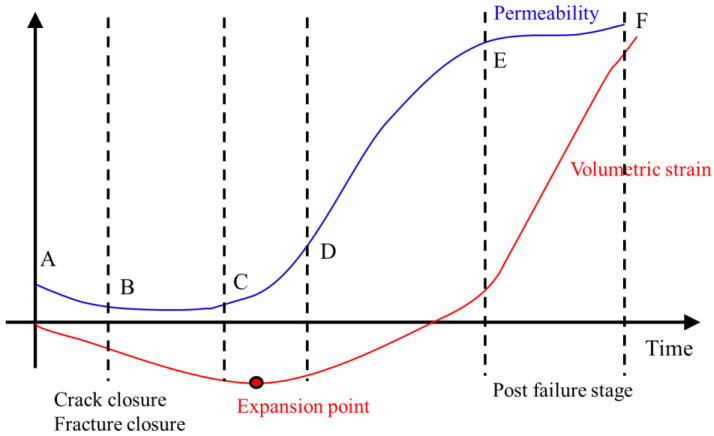
Schematic diagram for evolution of permeability and volumetric strain versus time.

**Figure 19 materials-15-08567-f019:**
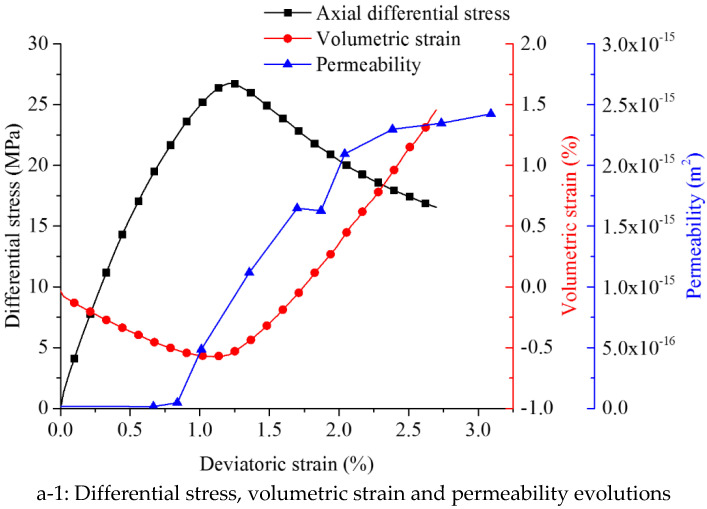
Permeability and strain evolution of coal sample C1 versus differential stress (numerical simulation results).

**Figure 20 materials-15-08567-f020:**
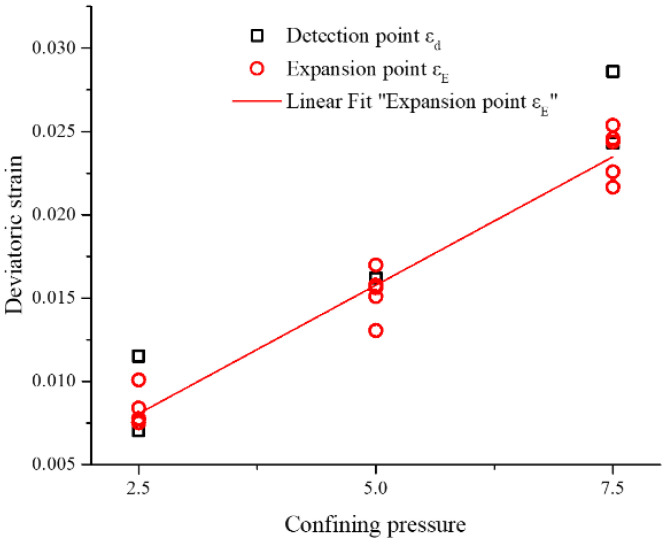
Deviatoric strain value *ε_d_* and *ε_E_* versus confining pressure.

**Figure 21 materials-15-08567-f021:**
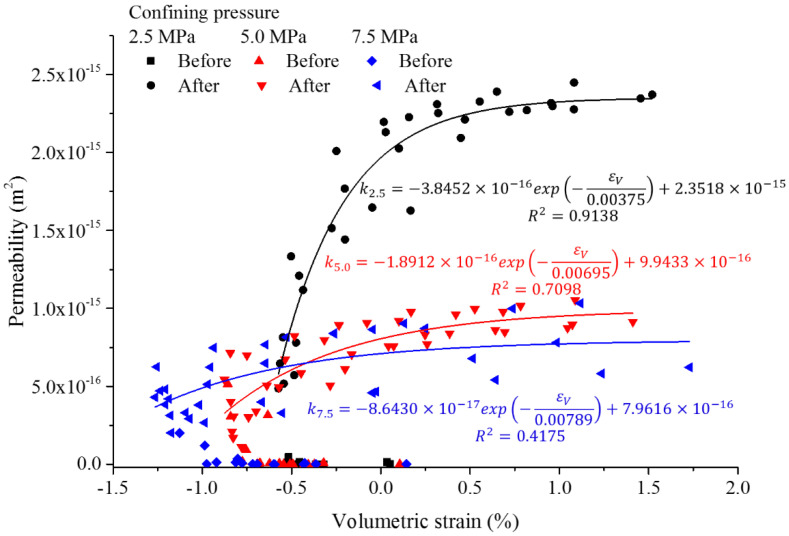
Permeability versus volumetric strain for different confining pressures obtained from numerical simulations (regression curves refer to values after the expansion point).

**Table 1 materials-15-08567-t001:** Composition of coal samples.

Sample No.	Proportions	
	Matrix	Inclusions
C1	80.12%	19.88%
C2	86.06%	13.94%
C3	79.06%	20.94%
C4	89.90%	10.10%
C5	90.20%	9.80%

**Table 2 materials-15-08567-t002:** Micro-mechanical parameters of the numerical coal model [18].

Element	Parameter	Coal Matrix	Mineral Inclusion	Boundary between Matrix and Inclusion
Particles	*N*	27,390	7218	--
	*r_p_* (mm)	0.5	0.5	--
	*ρ_p_* (kg/m^3^)	1390	1810	--
Contacts	$\bar{\lambda }$	1	1	1
	*E* (GPa)	2.09	2.84	1.90
	*K**	14.65	2.44	1
	$\bar{\sigma }$*_t_* (MPa)	17.40	30.50	11.60
	$\bar{c}$ (MPa)	4.35	12.20	5.8
	*μ_c_*	0.4	0.4	0.4
	*Φ* (°)	4	7	8
	*g_c_* (mm)	0.05	0.05	0.05
Axial loading control				
Walls	*V* (m/s)		0.005	
Confinement control				
Walls	*V_max_* (m/s)		0.01	

**Table 3 materials-15-08567-t003:** Mechanical and hydraulic parameters for numerical simulation.

Parameters	Matrix	Inclusions
*ρ* (kg/m^3^)	1500	2000
*E* (GPa)	1.5	3
*σ^t^* (Pa)	5 × 10^5^	5 × 10^5^
*C* (Pa)	2.7 × 10^6^	7.8 × 10^6^
*ψ* (°)	10	5
*θ* (°)	50	35
*ν*	0.35	0.35
*k_a_* (m^2^)	1 × 10^−18^	5 × 10^−16^
*K* (m/s)	8.75 × 10^−14^	4.38 × 10^−11^
*φ* (%)	6	9

**Table 4 materials-15-08567-t004:** Measured values of permeability (coal samples).

Sample No.	Confinement *σ*_3_ (MPa)	Dimensions		Peak Permeability *k_P_* (m^2^)
		Diameter (mm)	Height (mm)	
C1	2.5	25.50	50.60	2.8 × 10^−15^
C11	2.5	24.77	49.80	2.43 × 10^−15^
C13	5.0	24.99	50.02	8.85 × 10^−16^
C4	7.5	25.47	50.85	6.55 × 10^−16^
C12	7.5	24.89	49.89	2.67 × 10^−16^

**Table 5 materials-15-08567-t005:** Variables in Equation (14) for different confining pressures.

*σ_3_* (MPa)	*A* (×10^−16^)	*B* (×10^−16^)	*C*	*R* ^2^
2.5	−3.8452	23.5180	−0.00375	0.9138
5.0	−1.8912	9.9433	−0.00695	0.7098
7.5	−0.8643	7.9616	−0.00789	0.4175

## Data Availability

Not applicable.

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
