# Peer review of "Gas Permeability Evolution of Coal with Inclusions under Triaxial Compression-Lab Testing and Numerical Simulations"

_materials, 2022, doi:10.3390/ma15238567_

Round 1

Reviewer 1 Report

The article presents experimental and computational investigation of gas permeability in cracked / fractured coal samples under various levels of 3D compressive stress. The experimental setup seems to be appropriate and the procedure is technically sound. The equations provided appear to be adequate in order to assess the desired output. CT scans of the coal samples add value to the research study. The computational models are constructed properly and explained mostly in detail. The material properties assigned are thoroughly listed. The results generally appear to explain the phenomena and are in good agreement. The manuscript is well-structured and well-written. Command of English is mostly good. The authors are recommended to address the following comments in order to improve the quality of the paper:

1. In Section 4.1, the reasons for choosing parallel bonds and selecting a coefficient of friction of 0.1 should be elaborated. Is lubrication at the interface considered? If so, provide details.

2. In Figure 13.b for sample C11 the permeability trend is a decreasing one after a deviatoric strain of 0.02, which cannot be observed in a similar manner in the other samples (either a flat horizontal trend or an increasing one is observed.). Comment on the possible reasons for this behaviour for the sample C11.

3. In Figure 13.e, the permeability trend is a decreasing one for sample C12 between deviatoric strains of 0.00 and 0.01, which cannot be observed in a similar manner in the other samples (constant zero for the others). A comment would be useful.     

4. In various parts of the text, the same error message ("Error! Reference source was not found.") was observed several times regarding linking references to the relevant location within the text (lines 49, 224, 236, 253, 265, 266, 273, 349, 407, 438, 449). If this problem also exists in the original file, it should be sorted.

5. Some figures within the text were not properly pointed out; i.e. figure numbers were missing, such as in lines 162, 176, 338, 366, 

6. There is an abrupt gap in line 202.

Author Response

Response to Reviewer 1 Comments

  1. In Section 4.1, the reasons for choosing parallel bonds and selecting a coefficient of friction of 0.1 should be elaborated. Is lubrication at the interface considered? If so, provide details.

A: 1. A weak linear contact bond was originally assigned between particles and walls, this bonding is strong enough to provide a tight compression of the interface. 2. In this design, the surrounding walls are continuously changing their shapes and positions. The wall circle maintained closed and attached to the sample elements, confining pressures are applied in normal direction of each wall. The friction is only applied to prevent huge errors during wall moving phase, and its effects on simulation results can be neglected.

  1. In Figure 13.b for sample C11 the permeability trend is a decreasing one after a deviatoric strain of 0.02, which cannot be observed in a similar manner in the other samples (either a flat horizontal trend or an increasing one is observed.). Comment on the possible reasons for this behaviour for the sample C11.

A: explanations are added to the manuscript from Line 351-370.

  1. In Figure 13.e, the permeability trend is a decreasing one for sample C12 between deviatoric strains of 0.00 and 0.01, which cannot be observed in a similar manner in the other samples (constant zero for the others). A comment would be useful.     

A: explanations are added to the manuscript from Line 351-370.

  1. In various parts of the text, the same error message ("Error! Reference source was not found.") was observed several times regarding linking references to the relevant location within the text (lines 49, 224, 236, 253, 265, 266, 273, 349, 407, 438, 449). If this problem also exists in the original file, it should be sorted.

A: this problem is caused by a third-party tool. They are all fixed in the revised manuscript.

  1. Some figures within the text were not properly pointed out; i.e. figure numbers were missing, such as in lines 162, 176, 338, 366, 

A: this problem is caused by a third-party tool. They are all fixed in the revised manuscript.

  1. There is an abrupt gap in line 202.

A: this problem is caused by a third-party tool. They are all fixed in the revised manuscript.

Reviewer 2 Report

The paper is very interesting and it the topiuc is quite important because coal in mining in high depths is a world/wide problem.

However, there are some things which have to be improved. The style of text should be refined for easier readability (senteces  are sometimes too long - starting from the first sentence of the Abstract).

Please do not use things like "incl.", "hard-" and so.

There are many non-working references in the whole text (lines 49, 224, 236, 253, 265, 266, 438, 449).

The tests were executed at room temperature (20 degrees of Celsius). However, in the depths which are mentioned in introduction temperatures are much higher. Can the author explain if this fact can (or cannot) have influence on validity of their work?

Author Response

Response to Reviewer 2 Comments

The style of text should be refined for easier readability (senteces  are sometimes too long - starting from the first sentence of the Abstract).

A: the text of manuscript is modified according to this recommendation.

Please do not use things like "incl.", "hard-" and so.

A: such expressions are modified.

There are many non-working references in the whole text (lines 49, 224, 236, 253, 265, 266, 438, 449).

A: this problem was caused by a third-party tool. They are all fixed in the revised manuscript.

The tests were executed at room temperature (20 degrees of Celsius). However, in the depths which are mentioned in introduction temperatures are much higher. Can the author explain if this fact can (or cannot) have influence on validity of their work?

A: this work is mainly about a new simulation method based on continuum and discontinuum methods. The working target is to address the relationship between stress-induced fractures and general permeability. The temperature is not considered in this research. We totally agree, that temperature could have an influence on the considered phenomena. Our team is currently working on the introduction of temperature related parameters, the results will come in the future, but cannot be included in this paper.

Reviewer 3 Report

The paper presents a study on gas permeability evolution of coal with inclusions under triaxial compression – lab testing and numerical simulations

The following recommendations are proposed:

·         Please, improve the introduction section. It misses a comprehensive literature review.

·         Please, provide a flow chart of the paper organization.

·         Overall, English needs to be double-checked for typos.

·         The conclusions section can be better organized.

Main Concern:

What is the innovation that this paper brings into scientific knowledge?

There is a strong overlapping with another source.

Author Response

Response to Reviewer 3 Comments

  • Please, improve the introduction section. It misses a comprehensive literature review.

A: more relevant literature reviews have been added to the introduction.

  • Please, provide a flow chart of the paper organization.

A: a flowchart is added as Figure 1.

  • Overall, English needs to be double-checked for typos.

A: the text was carefully checked for typos.

  • The conclusions section can be better organized.

A: the conclusion section is modified.

Main Concern:

What is the innovation that this paper brings into scientific knowledge?

A: During exploitation of deep coal, the coal experience different loading stages. The variety of stresses leads to continuous deformation and permeability changes with development of internal pore-fracture networks. The internal pore-fracture network governs the permeability and influences the methane exploitation and mining safety.

Spatial distribution, dimension and hydraulic parameters of fractures are investigated on the basis of confined loading conditions. The evolution of the internal fracture network is investigated with DEM models. The development of microcracks is quantified by considering the loading, confinement and structural character of the rock samples. The simulatiosn focus on the permeability evolution under triaxial compression. The spatial distributions of different components inside the samples produce a significant heterogeneity. Precise reconstruction of the coal samples in form of numerical models is essential for better quantification of the structural influence on damage pattern and permeability evolution.

Therefore, deeper knowledge of these phenomena is the basis for the development of safe and effective exploitation strategies.

There is a strong overlapping with another source.

A: The existing paper “Damage Evolution of Coal with Inclusions Under Triaxial Compression” (from same Authors and Team) considers only the particle-based simulations and the corresponding construction method based on CT. It does not contain model set-up of the continuum code, no coupling with the continuum code, no flow modelling, no correlation between damage and flow pattern etc. Therefore, we believe the manuscript is a significant extension to the existing paper. 

Round 2

Reviewer 3 Report

The paper has a level of similarity of 48% (after removing the reference list), with 37% coming from a single source: tubaf.qucosa.de

The author should remove the similarity/overlapping.